# Invited perspectives: Safeguarding the usability and credibility of flood hazard and risk assessments

Bruno Merz[1,2], Günter Blöschl[3], Robert Jüpner[4], Heidi Kreibich[1], Kai Schröter[5], Sergiy Vorogushyn[1]

[1]Section Hydrology, Helmholtz Centre Potsdam, GFZ German Research Centre for Geosciences, 14473 Potsdam, Germany
[2]Institute for Environmental Sciences and Geography, University of Potsdam, 14476 Potsdam, Germany
[3]Institute of Hydraulic Engineering and Water Resources Management, Vienna University of Technology, Karlsplatz 13/223, 1040 Vienna, Austria
[4]Hydraulic Engineering and Water Management, Rheinland-Pfälzische Technische Universität, 67663 Kaiserslautern, Germany
[5]Hydrology and River Basin Management, Technical University Braunschweig, 38106 Braunschweig, Germany

*Correspondence to*: Bruno Merz (bmerz@gfz-potsdam.de)

**Abstract.** Flood hazard and risk assessments (FHRAs), and the underlying models, form the basis of decisions regarding flood mitigation and climate adaptation measures and are thus imperative for safeguarding communities from the devastating consequences of flood events. In this perspective paper, we discuss how FHRAs should be validated to be fit-for-purpose in order to optimally support decision-making. We argue that current validation approaches focus on technical issues, with insufficient consideration of the context in which decisions are made. To address this issue, we propose a novel validation framework for FHRAs, structured in a three-level hierarchy: process-based, outcome-based, and impact-based. Our framework adds crucial dimensions to current validation approaches, such as the need to understand the possible impacts on society when the assessment has large errors. It further emphasizes the essential role of stakeholder participation, objectivity, and verifiability when assessing flood hazard and risk. Using the example of flood emergency management, we discuss how the proposed framework can be implemented. Although we have developed the framework for flooding, our ideas are also applicable to assessing risk caused by other types of natural hazards.

## 1 Too little attention to the validation of flood hazard and risk assessments

Flood Hazard and Risk Assessments (FHRAs) play a pivotal role in flood design and mitigation (Sayers et al., 2016). They provide the foundation for informed decision-making regarding flood risk management. Decisions ranging from the design of flood protection infrastructure through spatial planning, developing flood insurance schemes, as well as emergency management and reconstruction after disastrous events rely on the information derived from FHRAs (Penning-Rowsell, 2015, Franco et al., 2020, Ferguson et al., 2023). The consequences of a flawed FHRA can be dire. These include but are not limited to: catastrophic economic losses, inadequate evacuation plans, erosion of public trust in governmental authorities, and inadequate flood control designs. Ensuring the usability and credibility of FHRAs is thus of paramount importance, particularly against the background of expected increases in flood risk (Merz et al., 2021, Wing et al., 2022).

At its core, FHRAs entail estimating flood hazard and risk and the effects of possible risk reduction measures through a structured way of thinking that involves the development of a model-based representation of a flood-prone area. In this

paper, we follow the widespread definition of risk as probability of adverse consequences (Merz, 2017). Risk is composed of hazard (the physical flood processes that may cause adverse impacts), exposure (people, assets, infrastructure etc. present in hazard zones that are subject to potential losses), and vulnerability (characteristics of the exposed elements that make them susceptible to the damaging effects of a flood). There is a wide spectrum of FHRAs in terms of: (1) the decision-making context, e.g. flood zone mapping, design of flood protection, or estimating insurance premiums, (2) flood types, e.g. river flooding, pluvial urban flooding, or flooding from dam failure, (3) impact, e.g. affected population, direct economic damage, interruption of services, (4) spatial and temporal scales, e.g. local, short-term assessments or global projections of future risk, and (4) depth of detail and available resources for developing an FHRA (de Moel et al., 2015, Sieg et al., 2023). Here, we discuss challenges related to the validation of FHRAs and propose a validation framework. We include all types of models for FHRAs: data-based statistical models, process simulation, event-based and continuous approaches, and on scales from local to global. We consider assessments that do provide complete risk estimates, such as assessments delineating failure scenarios and potential damage that result from flood events, quantifying the probability of their occurrence and estimating the associated consequences (Merz, 2017). But we also include assessments that are limited to hazard quantification. Examples for FHRAs include: urban inundation simulation using coupled models of catchment hydrology, river inundation and urban drainage system for scenarios of certain return periods (Jiang and Yu, 2022), large-scale assessments of current and future flood hazard utilizing a mixture of statistical and simulation tools (Bates et al., 2021), and process-based model chains that estimate the annual expected loss and other risk metrics on a national scale (Sairam et al., 2021).

The conventional understanding of model validation prevalent in hydrology and water resources management, but also in the broader field of environmental modelling, entails evaluating the alignment between a model and observed reality, such as streamflow observations (Biondi et al., 2012, Eker et al., 2018). Since the primary objective of an FHRA centres on facilitating decision-making processes, and because flood risk is not an observable phenomenon, we shift our emphasis from assessing the accuracy of a model vis-à-vis real-world observations to evaluating its ability to achieve its intended purpose. In essence, the evaluation of an FHRA model's validity is determined by its fitness for its intended purpose, reframing the criteria for assessment from correspondence to reality to alignment with decision-making needs. We thus define validation as "a process of evaluating a model's performance and suitability for its intended use" (Eker et al., 2018). While the term "validation" has been criticised for potentially implying that a model can be established as true, and for encouraging modellers to claim positive results (Oreskes et al., 1994, Wagener et al., 2022), we use the term validation because it is the preferred term for evaluating models in hydrology and water resources management. This understanding of validation embraces the notion of quality assurance defined as "the part of quality management focused on providing confidence that quality requirements will be fulfilled" (ISO 9000, 2005). Quality assurance is well established in areas such as industrial production. A quintessential objective of quality assurance is to attain a state of fit-for-purpose, wherein the product – the FHRA in our context – aligns with its intended application. Central to this notion is recognition that the end-user context plays a pivotal role. Quality assurance is, however, not limited to the evaluation of the end product but comprises also the

underlying processes. A key assumption is a strong relation between the quality of the underlying processes and the quality of the product.

Our focus on a fit-for-purpose approach follows earlier arguments. For instance, Lathrop and Ezell (2016) argue that, when discussing risk assessment for weapons of mass destruction, risk assessment is not a "risk meter," i.e. a device to measure risk. Instead, it constitutes a process of "reducing a large amount of information about a complex and uncertain situation into summary renditions targeted to supporting decisions." Sayers et al. (2016) argue similarly regarding the validation of flood risk assessments: "the process of structured reasoning about the level of confidence needed to support a particular decision and the credibility of the assessment of risk in that context." Outside the risk analysis literature, others have also argued for a broader approach to model validation than simply comparing simulation results with observations. For example, Eker et al. (2018) and Wagener et al. (2022) argue for more integrated approaches that also assess the conceptual and methodological validity, and Saltelli and Funtowicz (2014) propose sensitivity auditing, based on a 7-point checklist, to increase the credibility of a model. While most assessments attempt to validate flood hazard models (to some extent), validation of risk models has received little attention. Molinari et al. (2019) review the state-of-the-art of flood risk model validation based on two workshops with more than fifty experts. They conclude that "very few studies pay specific attention to the validation of flood risk estimates … Validation is perhaps the least practised activity in current flood risk research and flood risk assessment" (Molinari et al., 2019). This observation resonates with Goerlandt et al. (2017), who review validation in the more general field of Quantitative Risk Analysis. They summarize results of several benchmark studies in which various teams quantified the risk of a particular technical facility. In all studies, the teams produced widely varying numeric risk estimates, up to several orders of magnitude. Goerlandt et al. (2017) conclude that there has been little attention paid to the validation of Quantitative Risk Analyses.

This limited attention to FHRA validation is worrisome because disaster risk reduction and climate adaptation should be based on the best available information. Despite the shortcomings and (often substantial) uncertainty of FHRAs, they are extremely useful. The understanding gained through thorough risk analyses, awareness of the various ways a system might fail, and insights about the effectiveness of risk reduction measures constitute an enormous benefit to risk management. Against this background, this perspective discusses the challenges of validation for FHRAs (chapter 2), reviews what the science of model validation can offer (chapter 3), and proposes a validation framework for FHRAs (chapter 4). To illustrate how our framework aids in flood risk management, we provide an example from an important application – emergency flood management – in chapter 5. Our discussion will also be helpful for developing hazard and risk assessments for other natural hazards, since at present there is a general lack of attention to validation and quality assurance in risk research and safety science (Sadeghi and Goerlandt, 2021).

## 2 Challenges when validating FHRAs

### 2.1 FHRA validation as a messy problem

A wide range of challenges hampers the validation of FHRA. One fundamental problem is that flood risk, i.e. the probability distribution of damage, is not directly or fully observable. Extreme events that lead to damage are rare, and the relevant events may even be unrepeatable, such as the failure of a dam (Hall and Anderson, 2002). The rarity of extreme events results in a situation characterized by both limited data availability and increased data uncertainty. This uncertainty relates to data against which the flood model can be compared. For instance, streamflow gauges often fail during large floods, and losses are not systematically documented and reported losses are highly uncertain. In addition, input data is often insufficient for developing a viable flood model. For example, levee failures depend on highly heterogeneous soil properties, and levee-internal characteristics are typically unknown. Thus Molinari et al. (2019) conclude that "a paucity of observational data is the main constraint to model validation, so that reliability of flood risk models can hardly be assessed."

FHRA models, and their sub-models, can typically only be compared to rather frequent, observed events, and these models are then used to extrapolate to the range of extremes. This procedure raises a fundamental question: Are extreme floods the larger versions of more frequent floods? In many cases, the answer is no, because the mechanisms that lead to extreme floods differ fundamentally from those that lead to frequently occurring, smaller floods (Merz et al., 2021, Merz et al., 2022). Thus, even a well-calibrated model cannot be relied upon to predict or manage extreme events (Sayers et al., 2016). This extrapolation question relates to the completeness issue. Hazard and risk analyses should encompass all relevant scenarios that lead to damage (Kaplan and Garrick, 1981). But we may not be aware of all possible damage scenarios or failure modes, making an assessment limited to risks associated with the scenarios it lists and the processes it includes. Completeness thus constitutes a fundamental challenge for hazard and risk assessments (Lathrop and Ezell, 2016).

Another challenge concerns the complexity and non-stationarity of flood risk systems. These systems are affected by the complex interactions between human activity and natural processes at varying space-time scales, so that a system's behaviour can be hard to understand, quantify, and predict. Challenging examples include: upstream-downstream interactions (Vorogushyn et al., 2018), human-water feedbacks such as the levee effect (Barendrecht et al., 2017), and human behaviour affecting exposure and vulnerability (Aerts et al., 2018). The entire risk system may evolve over time, adding another issue for validation of FHRAs. A prominent example is the challenge of assessing the credibility of models that estimate flooding under climate change. A less obvious example is that risk can decrease rapidly following a disastrous flood, as people and institutions learn from the event and implement precautionary measures or retreat from hazardous zones (Bubeck et al., 2012, Kreibich et al., 2017).

Given these challenges, FHRAs are often strongly based on assumptions, expert judgement, and best guesses. FHRAs may thus be particularly prone to cognitive biases (biases in intuitive judgement) (Kahneman, 2011). Such biases may lead to overly optimistic estimates that neglect dramatic consequences or that lead to overconfidence, especially when assessors are unaware of the discrepancy between their perceptions and the actual risks (discussed in Merz et al., 2015).

## 2.2 FHRA decision contexts as complex landscape

FHRAs are needed for a wide range of decision-making contexts, but even specific situations involve multiple stakeholders with varying responsibilities and divergent perspectives. Stakeholders generally fall into three groups: risk analysts and modelers, decision makers, and people affected by the decisions. Risk analysts focus mainly on the scientific reliability of their FHRA, ensuring that it is based on state-of-the-art assessment methods: Are the right data used? Are all important processes included in the model? Decision makers are principally concerned with the usefulness of the FHRA, and whether the model's conclusions seem relevant to the system under consideration: Do the assessors and the public possess sufficient confidence in the FHRA to implement the decision? People affected by the decision are often particularly interested in their personal benefits and costs: Are my concerns considered? Are the costs and benefits fairly distributed? Finally, it should be noted that these three groups are rarely homogeneous, for instance, decisions may involve authorities from a variety of sectors with competing interests.

Decision makers and affected individuals must trust risk analysts and their modelling in order to take action based on the FHRA (Harper et al., 2021). Because they lack the expertise to assess the scientific validity of an FHRA, their trust in the results may depend less on scientific validity than on factors such as interpersonal relationships or the reputation of the risk analysts. Prior experiences with the analysts also play a role, since trust is self-reinforcing: trust breeds trust and distrust breeds distrust (Harper et al., 2021). As trust constitutes the outcome of a process in which a trusted relationship gradually evolves (Blomqvist, 1997), validation of FHRAs should be designed in a way that fosters trust. Lack of transparency undermines trust. For instance, in the last decade many (re)insurance companies have begun developing in-house risk models because commercial models tended to function as black boxes in which the underlying model assumptions and uncertainties were not transparent (Franco et al., 2020).

## 3 What the science of model validation can offer

### 3.1 Model validation in general

Model validation is relevant in many applications; Table 1 summarizes commonly utilized approaches. Several reviews of model validation have been published in the context of models in Earth sciences (Oreskes et al., 1994), hydrological modelling (Klemes, 1986, Biondi et al., 2012), ecological modelling (Aumann, 2007), models supporting environmental regulatory purposes (Holmes et al., 2009), computational models in biology (Patterson and Whelan, 2017), and terrorism risk models (Lathrop and Ezell, 2016). Some consensus emerges from this literature. Firstly, validation can establish legitimacy but not truth. Truth is unattainable because (geoscientific, economic, biological) systems are open, and input data are incompletely known (Oreskes et al., 1994). Models offer scientific hypotheses that cannot be verified but can be confirmed, for instance by laboratory or in-situ tests. Secondly, these authors agree that model validation must be carried out with a clear understanding of the purpose of the model. Finally, validation is a matter of degree, a value judgement within a

particular decision-making context. Validation therefore includes subjective choices (Holmes et al., 2009, Collier and Lambert, 2019).


**Table 1: Commonly used model validation approaches (based on Sargent, 2011; Harper et al., 2021)**

| Approach | Description |
| --- | --- |
| **Data validation** | Ensuring the adequacy and correctness of data used for model building, evaluation, and conducting model experiments. |
| **Historical data validation** | Comparing model results versus observed data, and quantifying whether the model performance is within prescribed limits. |
| **Predictive validation** | Using the model to predict the system's behavior and comparing the system's behavior and the model's predictions to assess whether model performance is within prescribed limits. |
| **Conceptual model validation** | Determining that the theories and assumptions underlying the model are correct, and that the model's representation of the problem and the model's structure, logic, and mathematical and causal relationships are reasonable for the intended purpose. |
| **Sensitivity analysis** | Changing the values of input and internal parameters, and/or the model structure, to determine the effects on the model's response. Identical relations should occur in both the model world and the real system. Highly sensitive parameters and model structure components should be made sufficiently accurate prior to using the model. |
| **Uncertainty analysis** | Providing uncertainty bounds for model results. Uncertainty can result from aleatory and epistemic uncertainty in model input or in model parameters and structure. Assessing whether the uncertainty bounds are sufficiently narrow for the intended purpose. Using knowledge gained from uncertainty analysis to improve models, e.g. including additional data for the most sensitive parameters in order to reduce uncertainty. |
| **Benchmarking** | Comparing model results to the results of other (independent) models to assess whether the model agrees sufficiently well with alternative models. |
| **Face validation** | Examination of model results by independent experts. Asking domain experts whether the model, its behavior, and results are reasonable for the intended purpose. |

**3.2 Model validation in FHRAs**

Using Table 1 as a guide, we provide a brief overview of approaches typically employed when validating FHRAs. Our overview complements the review by Molinari et al. (2019), which discusses the validation with respect to various components of flood risk. They conclude that some components of flood risk models are better validated than others. For hydrological and hydraulic models and flood frequency analysis, validation is often performed using observed streamflow data, water level data, and inundation data. Significantly less validation exists for model components whose data are scarce and whose mechanisms are difficult to quantify, such as flood defences: "historical data on flood defence failures are not enough for fully characterizing all potential failure mechanisms and the corresponding initiation and progression that lead to

flood defence failure" (Molinari et al., 2019). Another area with insufficient validation is the modelling of damage, especially in relation to indirect and intangible damage.

**Data validation** — ensuring that the data used in the risk analysis are appropriate and correct — is rarely discussed in an FHRA. One exception is the qualitative assessment of data uncertainties for the National Flood Risk Assessment (NaFRA) for England and Wales by Sayers et al. (2016). Data are generally treated as if they involved no uncertainty, even though they may contain significant errors and uncertainties. A recent example is provided by Sieg et al. (2023), who compare the asset values of businesses and residential buildings exposed to the 100-year flood areas in Germany using data from
OpenStreetMap and from land use based on the Basic European Asset Map (BEAM). The exposed business assets derived from BEAM are significantly higher. For Germany, the net asset values exposed to the 100-year flood areas using BEAM data are €366 billion for businesses and €191 billion for residential areas. The OSM data show exposed values of €92 billion for businesses and €176 billion for residential properties. The BEAM exposed values are thus 4.0 and 1.1 times higher than the OSM values. Such differences in exposure lead to corresponding differences in damage and risk estimates.

**Historical data validation** — testing how well the model compares with historical data — is the most common validation approach in FHRA and is typically used for hydrological models, hydraulic models, flood frequency models, and sometimes also for damage models (Schröter et al., 2014; Wagenaar et al., 2018). An obvious problem with this approach is that data are only available for a limited range of scenarios and return periods, so that the majority of the results fall into the extrapolation range. One of the rare comparisons of modelled risk, using EAD (Expected Annual Damage) as a proxy for
observed risk, is based on integrating 20 years of insured losses in the UK (Penning-Rowsell, 2021). The underlying assumption is that the observed EAD integrates a sufficient share of total risk. This comparison finds that modelled flood risk at the national scale is between 2.1 and 9.1 times the corresponding flood loss measured in terms of the insurance compensation paid. In contrast, Bates et al. (2023) find a very good agreement (difference of 2%) between the simulated EAD for 2020 for UK with the observed value reported by the Association of British Insurers. Sairam et al. (2021) compare
the simulated damage for large-scale flood events in Germany between 1990 and 2003 with reported damage; for four out of the five events, the uncertainty bounds encompass the reported damage. The damage of the event in 2002 is substantially underestimated by the model, which can be explained by the more than 100 dike breaches not considered in the model. Little data exist to aid in assessing processes such as dam or levee breaches or the behaviour of humans in flood situations. Sometimes, deficits in individual model components are compensated by adjusting other model components where data are
scarce; for instance, by adjusting vulnerability functions so that damage estimates agree with reported values (Déroche, 2023).

The ideal model-building process utilizes an initial model to make testable predictions, then takes measurements to test and improve it (Ewing et al., 1999). This **predictive validation** approach appeals because the modeler is unaware of the measurements at the time of the model experiment and is therefore not subject to hindsight bias. Nonetheless, predictive
validation requires that relevant events occur while the risk assessment is performed; a condition rarely met in an FHRA. However, one could withhold some observations and compare varying assumptions or models with the withheld

observations (Holländer et al., 2009). In that way, the modeler operates in a situation similar to that of predictive validation, and the validation process would improve on the typical situation in which more effort is spent on the refined estimation of model parameters than on a thorough understanding of the mechanisms (Hölzel et al., 2011).

The aim of **conceptual model validation** is to ensure that the right outputs are produced for the right reasons (Biondi et al., 2012). A conceptual model represents an abstraction from the real-world system under study. Developing a conceptual model requires identifying what to include in the model and what to omit, and choosing the appropriate level of simplification. The basis for selecting a particular conceptual model is that it represents mechanisms and features of the real system considered essential, and is consistent with observations and general principles. Criteria, such as computational

simplicity or familiarity with the model, should not play a role (Ewing et al., 1999). However, deciding how to perform a conceptual model validation is more difficult in practice (Biondi et al., 2012). Conceptual model validation is rarely mentioned in FHRAs. Sayers et al. (2016) propose mapping out important real-world processes and model-world processes to ensure crucial processes are neither ignored nor misrepresented. For the national FHRA for England and Wales they list model simplifications and discuss how these might affect results.

**Sensitivity and uncertainty analysis** have received widespread attention in flood hazard studies (Savage et al., 2016, Beven et al., 2018, Hu et al., 2020, Xing et al., 2021) and flood risk studies (Tate et al., 2015, Bermúdez and Zischg, 2018, Rözer et al., 2019, Sairam et al., 2021). For example, Merz and Thieken (2009) implement uncertainty analysis to ascertain dominant sources of uncertainty and how these change with return periods. Similarly, research in the UK reveals that uncertainty in fragility curves — data that represent the probability of flood defence failure — may impact the basic estimate of risk (here:

expected annual damage) by a factor of between 0.5 and 2 (EA, 2002). An advantage of sensitivity and uncertainty analyses is that they offer a range of established methods and tools (Beven and Hall, 2014, Pianosi et al., 2016, Page et al., 2023) that can be applied without observational data. Still, this approach encounters problems. Sensitivity and uncertainty analyses only examine sensitivities and uncertainties within the conceptual model chosen and are blind to omitted concepts and processes, as Ewing et al. (1999) state: "Solving the wrong equations (i.e., using a flawed or erroneous conceptual model) many times

based on a wide range of parameter values will not necessarily yield a meaningful probabilistic distribution of outcomes." There are proposals to take a broader view of sensitivity analysis in model evaluation that go beyond modelled input-output relationships (response surfaces). For example, Wagener et al. (2022) argue that (global) sensitivity analysis can provide transparency of model behaviour and increase stakeholder confidence in model results, particularly in data-poor situations, by addressing a range of questions such as: Are the modelled dominant process controls consistent with our perception of the

system?

**Benchmarking** — comparing the model with alternative models (or alternative model components) — occurs rarely in the field of FHRAs. Existing benchmarking studies tend to focus on methodological issues to understand which methods perform well for a given (benchmark) dataset (e.g. Matgen et al., 2011 for benchmarking methods for delineating flooded areas from SAR data). Other studies, e.g. Bates et al. (2023), focus on benchmarking the resulting risk metrics, in this case

EAD from flooding in the UK, between different approaches without a detailed comparison of the models. The enormous

effort that is often required to implement flood models and to harmonize them in order to allow comparison explains the lack of benchmarking in FHRA. Including several independent models is an effort considered prohibitively expensive for most applications. Such efforts are more easily justified on large spatial scales. For instance, Trigg et al. (2016) compared the results of six global flood models for Africa.

**Face validation** — consulting independent experts to judge the credibility of an FHRA — is rarely done, at least within a systematic expert consultation process. Lamb et al. (2017) relied on expert elicitation workshops to estimate the vulnerability of bridges to scour during flooding. However, their purpose was not to validate an FHRA, but to obtain risk estimates in a situation where models were unavailable.

This overview demonstrates that validation approaches have been used to varying degrees in FHRAs. The most common
strategies are historical data validation—albeit mostly for the hazard component of flood models, and sensitivity and uncertainty analysis. Others are rarely used or unreported in the scientific literature.

## 4 Matching validation with the decision-making context

Models are representations of the real world used to understand the system under study and to reveal its behaviour in situations that provide little or no observational data. Flood models have an additional dimension the moment they are
intended to inform decision-making. For example, the costs of optimism differ from those of pessimism (Begueria, 2006). False negatives (the model does not simulate a flood in a situation when a flood does occur) are worse than false positives (the model simulates a flood where a flood does not occur). In the latter case, the consequences may involve lower cost-benefit ratios than assumed or, in the extreme case, the uselessness of mitigation measures. False negatives, on the other hand, may lead to the destruction of buildings and infrastructure, even loss of life. The appropriate level of optimism
depends on the specific decision-making context. Embedding the validation of an FHRA in a specific decision-making context requires that stakeholders' perceptions and concerns are considered and that conflicts can be resolved (IRGC, 2017).

### 4.1 Novel framework for a decision-sensitive validation of FHRAs

The literature concerning the validation of FHRAs focuses on methodological issues rather than on comprehensively considering the specific decision-making context. We propose a novel framework that enhances current validation
approaches by addressing both challenges outlined in Section 2: the messiness of FHRA validation and the complex decision-making contexts of FHRAs. Similar to Carr et al. (2012), who develop a three-level scheme to evaluate the participation in water resources management, our framework consists of three levels (Table 2). Level 1 (procedure-based) ensures that the design and organization of the FHRA is transparent, documented, and well-embedded in the specific decision-making context. Level 2 (outcome-based) provides information regarding the level of confidence in the outcome of
the FHRA. Level 3 (impact-based) considers how harmful the decision could be were the FHRA inaccurate. These seven criteria are discussed in greater detail below.

**Table 2: Proposed framework for the validation of FHRAs**

| Criterion | Description |
|---|---|
| *Level 1: Procedure-based* | |
| **Participation** | Relevant stakeholders are involved, agree on the decision-relevant aspects to be analyzed, and express their concerns and perspectives. |
| **Objectivity** | The FHRA is unbiased by personal views and agreeable to most peers. |
| **Verifiability** | The FHRA is transparent and reproducible. Peers and users can understand the FHRA's conceptual basis, assumptions and uncertainties. |
| *Level 2: Outcome-based* | |
| **Accuracy** | Differences between model results and real-world data are given. |
| **Precision** | Uncertainty bounds of model results are provided. |
| **Gross Error Potential** (GEP) | Potential for major errors that could lead to wrong risk estimates is considered. |
| *Level 3: Impact-based* | |
| **Consequentiality** | Consequences of errors, gaps, and uncertainties in the FHRA on the decision are considered. |


The **participation** of all relevant stakeholders in an FHRA is essential because they offer indispensable local/regional knowledge and insights, and their competing concerns and perspectives contribute to the quality of an FHRA and the resulting management decisions (Bähler et al., 2001, IRGC, 2017). Because FHRAs involve ethical judgments — for example in determining which types of damages to include (or omit) in the assessment (Fischhoff, 2015) — involving

stakeholders in a two-way communication process increases the effectiveness and fairness of an FHRA and the likeliness of stakeholders accepting the decisions made by governmental agencies (IRGC, 2017). Participation can address the erosion of credibility likely to occur in situations with highly uncertain information (Doyle et al., 2019). Broad participation also aids in determining the appropriate level of detail of the FHRA. Because increasing the level of detail (e.g. in terms of processes involved, increased resolution, and uncertainties considered) rapidly increases effort and cost, the level of detail should be

tailored to the decision-making context.

Designing and performing an FHRA involves scientific judgements, which can substantially influence the results and risk reduction decisions (Sieg et al., 2023). Therefore, the criterion **objectivity** aims to lead to an FHRA that is acceptable to most parties and largely unbiased by personal views, although no FHRA can be completely objective (Viceconti, 2011).

FHRAs should be documented and presented in a way that allows both decision-makers and affected people to understand

their conceptual basis, underlying assumptions, and uncertainties (Viceconti, 2011). FHRAs should be transparent and reproducible. **Verifiability** is an essential basis for achieving participation and objectivity.

FHRAs should clearly quantify and present their **accuracy** — the differences between model results and observations for as wide a range of data as possible. As the final result (e.g. the 100-year flood map or the risk curve) is difficult to validate due to its probabilistic nature and the rarity of extreme events, all sub-models should be compared to observed data whenever possible. Validating each component of the flood model reduces erroneous risk statements and error compensation, i.e. adjusting the parameters of one model component to compensate for deficits in others (Aumann, 2007).

Each FHRA should quantify its **precision** by providing a statement regarding uncertainty, because uncertainty is a key factor in the decision-making process (Downton et al., 2005, Doyle et al., 2019). Palmer (2000) demonstrates that probabilistic forecasts of weather and climate have greater potential economic value than single deterministic forecasts troubled by an indeterminate degree of accuracy.

The **Gross Error Potential (GEP)** is defined as the potential for a major or fundamental mistake in an FHRA that may lead to (very) wrong risk estimates (Thoft-Christensen and Baker, 2012, Sayers et al., 2016). Examples of GEPs include: unrecognized yet significant failure modes, or important interactions between processes or components that have not been considered. GEP relates to the criteria accuracy and precision, but takes a wider perspective. The former two are quantitative and linked to the selected assumptions and models. For instance, precision is often quantified by an uncertainty interval resulting from a range of plausible parameters. In contrast, GEP is a more general (and often qualitative) reflection on errors that could occur were the selected assumptions and models wrong. For example, we might not be able to include the mobilization, transport, and deposition of sediments, debris, and deadwood in a flood model; but still it might be important to reflect whether this simplification might lead to a large error in our model results.

**Consequentiality** relates to the harm that could follow from errors and uncertainties in the FHRA: What might be the consequences if the FHRA is wrong or affected by great uncertainty? The FHRA must be evaluated by the harmful consequences of its (known or potential) errors according to the decision-makers' perspectives. Ben-Haim (2012, p. 1644) calls this the "model robustness question." For instance, what would be the consequences if the model was off by one order of magnitude?

## 4.2 How to apply our framework in FHRAs

In order to animate our framework, we discuss the elements of validation that contribute to the seven criteria of Table 2. We take these elements from model validation literature and include insights from literature on risk governance, decision making, and stakeholder participation. Figure 1 visualizes how the various elements affect the criteria of our framework.

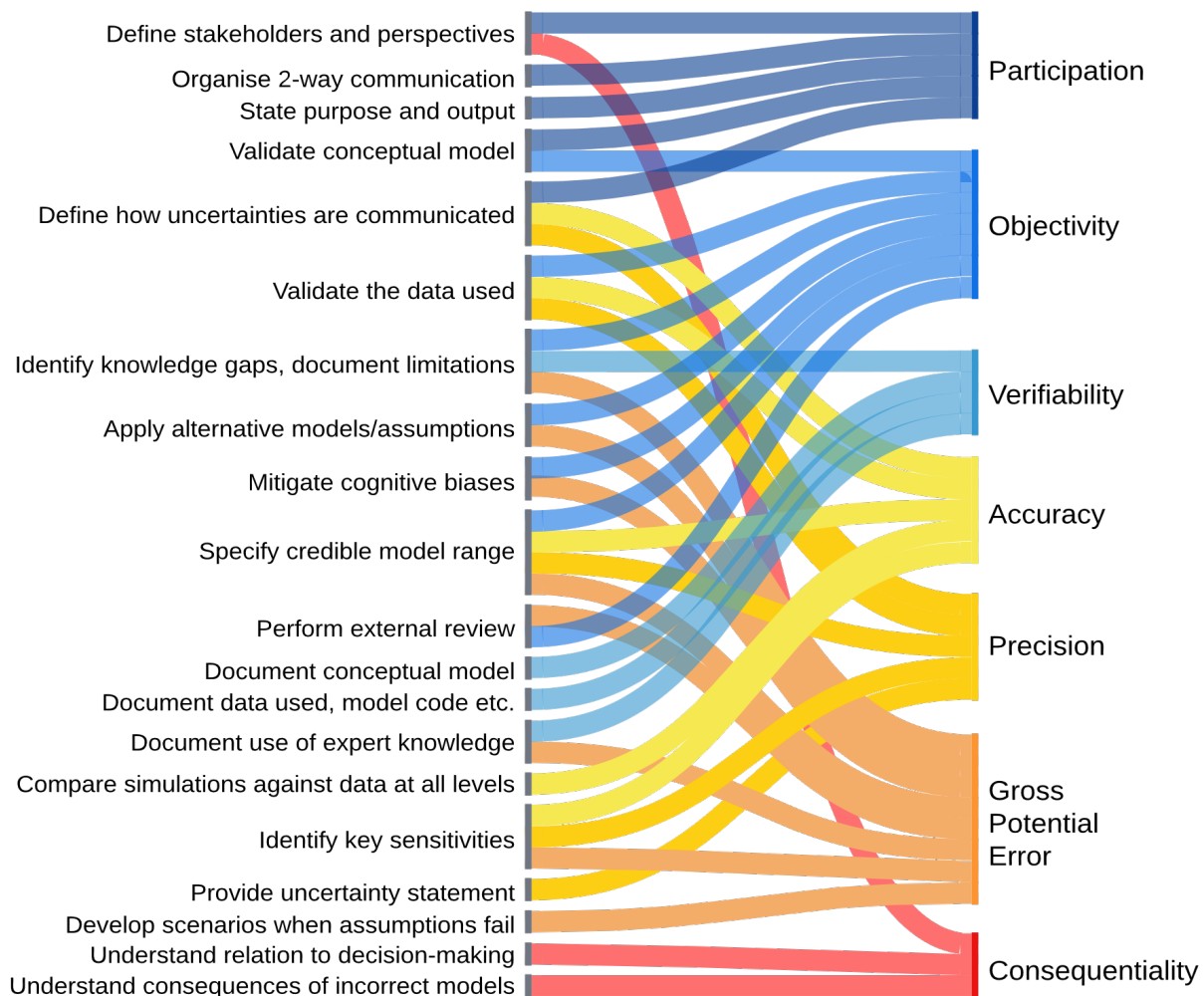

Figure 1: Validation elements (left) and their assignment to the criteria (right) of the decision-sensitive framework for the validation of FHRAs

**Define stakeholders and their perspectives**: Stakeholders (affected people/risk takers, decision makers/risk managers, modellers/risk analysts) and their perspectives must be identified (IRGC, 2017): What are their views and concerns regarding risk and possible risk-reduction measures? Decision-makers should understand how the FHRA's design in terms of scope, scale, and temporal horizon affects various risk takers. For example, might conflicts arise due to the unequal distribution of risks and benefits?

**Organise two-way communication**: A structured procedure for communication among risk analysts, decision makers, and affected people should be organized (Patt, 2009, Dietz, 2013, IRGC, 2017). This includes the internal communication process between risk analysts and decision makers, as well as external communication with the public, especially flood-

prone citizens (IRGC, 2017). Effective communication plays a pivotal role in establishing trust in risk management (IRGC, 2017). Research shows the importance of respectful relationships where stakeholders are involved in defining and interpreting the analyses (Fischhoff, 2015).

**State the purpose of an FHRA and the required output**: Each FHRA should be accompanied by a clearly formulated 'intended use statement' (Elele, 2009) and a description of its results, such as the types of damage and risk metrics included or the level of confidence required. This statement should ensure that all decision-relevant issues are adequately addressed (Sayers et al., 2016). For example, a regional FHRA that explores large-scale climate adaptation will have very different requirements than a local assessment undertaken in order to design flood protection for an industry that may release hazardous substances if flooded.

**Validate conceptional model**: Careful analysis of the processes and relationships that contribute to flood risk and how they are represented in the FHRA helps ensure that important processes and relationships are not ignored or misrepresented (Sayers et al., 2016). This discussion should include justification for the chosen system boundaries, since those of modelled systems are typically much narrower than in reality. Some examples of processes often excluded in FHRAs but which can have significant impact on flood risk include: contamination of flood water (Thieken et al., 2005), compound events such as the combined effects of river floods and coastal surges (Zscheischler et al., 2020), or upstream-downstream interactions along rivers (Farrag et al., 2022).

**Define how uncertainties are communicated**: Uncertainties embedded in an FHRA can be large, and uncertainty statements are often difficult for stakeholders to understand. Since disclosure of uncertainties does not always increase trust and credibility in risk analyses (Doyle et al., 2019), information about them should be tailored to the specific audience and consider their perspectives, technical knowledge, and concerns (Loucks, 2002, Faulkner et al., 2007, Beven et al., 2018).

**Validate the data used**: FHRAs should include data validation because it is essential to the creation of accurate models. This includes all types of data: observational, simulated, literature data or data based on expert judgement. Fundamental data gaps and strategies for compensating for them should be identified and justified.

**Identify knowledge gaps and document model limitations**: Knowledge gaps, simplifying assumptions, and model limitations should be discussed in the same detail as model strengths (Biondi et al., 2012). This discussion should include those cases in which the model does not obtain satisfactory results and possible reasons for such results. Discussion of fundamental knowledge gaps, how these are addressed, and whether they may lead to large errors should be part of an FHRA.

**Apply alternative models/assumptions**: Alternative hypotheses regarding the conceptual or mathematical models chosen and the reasons why other alternatives were not selected should be discussed (Biondi et al 2012). If resources allow, multiple alternative models should be generated, as described in Chamberlin's 'Method of Multiple Working Hypotheses' (Chamberlin, 1980).

**Mitigate cognitive biases**: Flood risk systems have characteristics that can easily lead to cognitive biases (Merz et al., 2015). The literature on cognitive biases (Kahneman, 2011) and 'High Reliability Organizations' (organizations that operate

in high-risk domains without serious accidents or catastrophic failures, such as the aviation industry, see Weick et al., 1999) offers strategies to mitigate cognitive biases. Although these biases cannot be eliminated completely, awareness of and efforts to address them help to reduce biases in an FHRA.

**Specify range for which model is validated/credible**: No model remains credible for all possible situations (Viceconti, 2011). It is therefore important to state the range for which the model is credible, i.e. for which we assume that the model provides information on the real-world system behaviour that is sufficient for the decision context at hand. To this end, the range of return periods, failure mechanisms etc., for which data exist, should be specified, as should those cases for which observations are unavailable. One should also try to answer the extrapolation question — whether extreme events are the large-scale version of frequent events. Might varying processes dominate, or might particular processes change behaviour between the observational and the extrapolation domains?

**Perform external review**: In areas such as the aerospace industry, formal external reviews for simulation models have been developed through a certification process (Kaizer, 2015), and maturity assessment frameworks have been developed to assess whether a simulation should be trusted for its intended purpose (NRC, 2007). Formal external reviews help to achieve objectivity and detect errors.

**Document conceptual model including scientific and ethical judgements**: To promote understanding and acceptance of an FHRA, its conceptual model and the scientific and ethical judgments made in designing and conducting the analyses must be documented, disclosed, and justified. This relates to the system boundaries chosen, the processes included or omitted, their representation in the risk model, and the assumptions made (Sayers et al., 2016).

**Document data used, model code, model workflows and calibration etc**.: To achieve reproducibility, all important pieces of information about the FHRA must be documented. This includes the model code, the data used, details of the computational environment (e.g. package versions), workflows, use of version control software (Hall et al., 2022), and protocols documenting the calibration and validation of the FHRA.

**Document use of expert judgement**: Expert opinions often inform an FHRA, for example, to determine parameter values or uncertainty intervals. In most cases, expert judgement is incorporated informally into the assessment, rather than through structured expert discussions where judgements are transparent (Lamb et al., 2017). This makes it all the more important to identify the use of expert judgement. Facts should be distinguished from opinions and best guesses (Mohanty and Sagar, 2002). Given the prevalence of over-confidence and other cognitive biases, it is important to be clear when expert judgement is involved.

**Compare simulations against data at all levels**: When comparing model results with data, sub-models and the overall flood model should be compared as extensively as possible. Even a good performance at the overall system level is no guarantee that sub-models are correct. When model errors can be assessed against historical data, they should be checked for consistency, stationarity and residual structure (Beven et al., 2018).

**Identify key sensitivities**: Key sensitivities, such as those arising from model assumptions and influential parameters or datasets, should be identified and documented. Particular attention should be paid to the issue of heavy tails in probability distributions and tail dependence (Beven et al., 2018).

**Provide an uncertainty statement; list uncertainty sources not included**: Ideally, a formal uncertainty assessment would include all sources of uncertainty (data, model structure, parameter). To understand how comprehensive the uncertainty assessment is, it should be disclosed where important sources of uncertainty are not addressed and explain how this might affect the confidence in the results of an FHRA.

**Develop scenarios under which an assumption fails**: One of the main objectives of an FHRA is to uncover hidden dangers

and reduce the potential for adverse surprises for flood-prone individuals and risk managers. To understand the gross potential error, risk analysts should determine what consequences might unfold when key assumptions fail.

**Understand relationship between FHRA and decision-making**: Validation of FHRAs is a matter of degree (Collier and Lambert, 2019), because flood models cannot fully be confirmed through observation, and few models can completely be refuted (Aumann, 2007). Model validation therefore involves a judgement about the appropriateness of a model for the

purpose of an FHRA. With this in mind, it is essential to understand the extent to which decisions such as the design of risk reduction measures depend on an FHRA (Elele, 2009). Clarifying the role of an FHRA within the decision-making context helps to determine the effort to invest and the level of confidence to be achieved.

**Understand consequences of incorrect models**: When an FHRA is instrumental for a particular risk management decision, it is important to understand the consequences of an incorrect or overly optimistic model. Is the situation at hand in terra

benigna or terra maligna (Merz et al., 2015)? In the first case, a poor decision might, for example, reduce the benefit-cost ratio, but would not be disastrous. In the second case, a poor decision might have disastrous consequences, such as destruction of infrastructure or loss of life.

Currently, many elements that can support the validation of FRAs (Figure 1) are either ignored or not explicitly considered. Applying all validation elements could easily overwhelm an FHRA, and in many instances the additional insights gained by

applying all of them would not justify the additional cost. However, it seems evident that current approaches to validating FHRAs lack crucial considerations and need to be broadened. We feel strongly that past emphasis on technical issues in FHRA validation needs to be reassessed and complemented by efforts to ensure the quality of the validation process (procedure-based criteria in Table 2) and to understand the consequences of possible errors (impact-based criterion in Table 2). Our list of validation elements is thus meant as an appeal to reflect on the elements most useful to an effective validation

process.

These elements are not intended as a recipe to be strictly followed, and our framework does not answer the question of how to decide on the appropriate thresholds that define that a FHRA is "sufficiently valid". This is beyond the scope of this perspective, as the specific thresholds and the ways to decide on them will vary between different contexts. We believe that our framework helps to decide whether the specific model is suitable for a given context. We follow Howard (2007) who

discussed the related problem of what constitutes a good decision. According to Howard, a decision should not be judged

strictly by its outcome, as a good decision does not always lead to a good outcome, and a bad decision does not always lead to a bad outcome. Instead, a good decision is determined by the process by which one arrives at a course of action. Howard then defined six (decision quality) elements and argued that good decisions are those in which all of these elements are strong. Similarly, our framework can support the discussion of the degree of validation of a model. We believe that good

validation is determined by the process used to evaluate a model. Applying our framework, we argue that a particular model is validated when all seven criteria are met to an extent that is appropriate in the specific context.

In the process of model validation, there is usually a decision that the specific model is "sufficiently valid" (despite its less than perfect performance). Such decisions are needed because a flood protection measure requires design characteristics, or

any other real-world decision has to be made, based on a concrete model output, which can be a single number, a probability distribution, a set of what-if scenarios, etc. In such a situation, one could consider the degree to which the model is valid in the decision context. For instance, reliability engineering (e.g. Tung, 2011) considers (aleatory and epistemic) uncertainty in the design of structures. In a situation where the available model is less able to reproduce the observations, one can consider this lower validity in a wider probability distribution of the load (external forces or demands) on the system or the resistance

(strength, capacity, or supply) of the system. This, in turn, will lead to higher design values due to the high uncertainty represented in the specific model. In a situation where one has several alternative models, each associated with a measure of its validity, one can weight these models to obtain the concrete model output required for a specific decision.

In summary, our framework is an attempt to address the challenges of FHRA validation, including the need for validation without data. Our framework goes beyond the current, most prominent view of model validation (which is strongly focused

on historical data validation, i.e. on comparing simulations and observations) by adding validation elements and criteria (see Figure 1) that can be applied without observations.

**5 Considerations for emergency flood management**

To illustrate how our framework adds value to current practice, we outline how it can be applied to emergency flood management. We chose this application deliberately, because emergency management tends to act on tacit knowledge

without relying on systematic FHRAs. Water infrastructure design, regional planning, and the insurance sector base their decisions on FHRA models to a much greater extent. We thus focus on the sector that is furthest away from flood modelling and comprehensive FHRAs and is strongly challenged by the need for validation without data.

We can distinguish two modes of emergency flood management: routine and for extraordinary situations. In many countries, levees and other flood defences are designed to cope with a 100-year flood, and few events reach or surpass flood defences.

In these cases, emergency management focuses on routine measures as part of an 'alert and operations plan'. If the water level at a gauging station exceeds a predefined threshold, emergency measures will be triggered automatically. This may

take a variety of forms: the closing of a road susceptible to flooding or the erection of a temporary protection wall. The responsibility for coping with such floods falls on municipal firefighters or other local emergency forces.

Recent catastrophic floods, such as the 2013 Elbe floods in Central Europe, the July 2021 floods in Western Europe, or Storm Daniel in September 2023, which devastated the Mediterranean region including Greece, Turkey, and Libya, have highlighted the need to be prepared for extreme floods. For emergency management forces, such extreme events are exceptional situations and knowledge of routine measures is insufficient to successfully mitigate the impending disaster. The Elbe flood was typical of the complex web of challenges that arise during an extreme event. On 10 June 2013 around midnight, the Elbe River levee broke, rapidly opening to a width of 90 m (Dagher et. al., 2016). Five days later, 227 million m³ of water inundated 150 km2 and several thousand people had to be evacuated (LHW, 2016). This was an unexpected development for the unprepared emergency management forces. What made matter worse was the lack of a flood model for calculating the spatial distribution of the flood water and how that would unfold over time, not to mention the lack of a strategy for closing the levee break.

In case of a situation as complex as a catastrophic flood event, good decision-making requires suitable flood forecasts as well as the ability to explore potential flood scenarios, their impacts, and the effectiveness of emergency measures. Flood models can contribute to improving emergency management in such cases. Firstly, they could be used for training and education of (local) emergency managers, who could be organized into a 'rapid response team' linked to a professional emergency management organization such as the Federal Agency for Technical Relief in Germany. Establishing emergency management forces with knowledge on flood risk management would be a tremendous aid to persons in charge during extraordinary flood situations. Secondly, flood scenarios could be prepared in advance and stored in a knowledge base. Modern modelling and software tools can generate various scenarios, including failure of flood defences and consequences for flooded assets, which can be used during an impending event to support emergency management. In an even more advanced setup, flood models could be run as soon as an extraordinary situation is expected to develop, incorporating near-real time information, for instance, on the state of levees. Table 3 outlines how our framework would support the decision-making of emergency managers.

*Table 3: Applying the proposed validation framework to an FHRA to support emergency flood management. For the sake of brevity, only a few validation elements are selected and presented.*

| Context of FHRA: Development of an FHRA tool to be used by emergency managers to decide on emergency actions. As soon as the official flood forecasts for a region indicate that inundation and losses could occur, emergency managers should be able to run the tool. It should allow simulating in near-real time flood scenarios, including failure of defenses, and quantifying who and what would be affected, as well as when and how. Based on these scenarios, emergency managers can decide on actions to minimize the adverse consequences. |
| --- |

| Validation element | Description |
| --- | --- |
| Define stakeholders and their perspectives | Emergency managers and their demands and concerns are identified. Examples include: decision structure, technical knowledge in terms of flood processes, levee safety or evacuation options, understanding the degree of uncertainty provided by the flood forecast and flood tool. Affected populations and their demands and concerns are identified. Examples include: support needed in case of evacuation and unequal consequences of emergency measures, e.g. intended breach of levees, for various areas. |
| State purpose of FHRA and output required | An 'intended use statement' is formulated that describes the use cases (e.g. when will the tool be applied? What is the time available for decision-making? Who will apply the tool?), the input (e.g. ensemble flood forecasts at certain river gauges for varying lead times) and the output that can be generated (e.g. hazard indicators such as water depths, flow velocity, time to inundation; assets considered in inundation scenarios such as affected people, affected buildings, and objects of sensitive and critical infrastructure). |
| Document conceptual model, including scientific and ethical judgements | The system boundaries chosen, the processes included or omitted, their representation in the scenario tool, and the assumptions made are documented and justified. Emergency managers and affected population have access to the conceptional model and its justification (e.g. why certain impacts are not included in the tool). |
| Validate conceptional model | A discussion with emergency managers, local experts, and affected population ensures that important processes (e.g. levee breaches) and loss-influencing factors (e.g. temporal variability in the number of flood-prone people, potential sources of contamination during flooding) are considered. |
| Identify knowledge gaps and document model limitations | Knowledge gaps are discussed such as levee breaching (e.g. the timing, location, width of breaching cannot be forecasted), indirect impacts of inundation (e.g. the vulnerability of power supply and water provision is unclear), effectiveness of emergency measures (e.g. the success of evacuation cannot reliably be predicted). |
| Identify key sensitivities | Key sensitivities are investigated and transparently described for emergency managers. These include the sensitivity of the output (e.g. time to inundation at certain locations) to the flood forecasts, to simulations of levee breaching and of inundation, and the key assumptions involved. |
| Develop scenarios under which an assumption fails | The conceptual model and assumptions are screened and potential errors discussed. Examples include: a levee failing even though the river level is well below the levee crest; bridges clogged with debris causing localized increases in river levels that might lead to flooding not represented by the tool. |
| Understand consequences of incorrect models | The consequences of emergency decisions, that result from using the flood tool, on the affected population and their assets are assessed are reflected upon. Examples include: tool falsely simulates levee failure and thus underestimates the rapidity and degree of inundation; might such a case endanger the evacuation? |

The use case of emergency flood management exemplified in Table 3 reflects the situation that flood models for the management of extraordinary emergency situations cannot rely on the typical element of validation, namely comparing model simulations with observed data. Only rarely does observed data about inundation, defence failures, and impacts exist for a particular region. This is precisely why it is all the more important to safeguard the usability and credibility of the models applied by complementing the traditional approach with a largely observation-independent approach. The latter includes elements that ensure a robust validation procedure (level 1 of the framework in Table 1: procedure-based), so that the assessment is unbiased, transparent, consistent with the existing knowledge and relevant to the decision context. It also includes elements that ensure that the sensitivity and uncertainty of the model are fully understood, including the potential occurrence of major errors (level 2: outcome-based). Finally, it tries to clarify how serious possible errors are (level 3: impact-based). Such an approach could, for example, lead to a decision to evacuate people even though the probability of life-threatening flooding is low given the uncertainty in the current flood forecast.

## 6 Conclusions

The validation of FHRAs is a neglected topic in the scientific literature, and presumably also in practice. This is problematic because, firstly, FHRAs rely heavily on assumptions, expert judgement, and extrapolation and, secondly, incorrect FHRAs can have catastrophic consequences. We believe the current practice of FHRA validation lacks essential elements. We make several proposals with this in mind: Stakeholder engagement, objectivity, and verifiability, but also the consequences of model errors on affected people need more attention. This perspective paper is intended as a clarion call to the community of flood experts in science and practice to reflect on the shortcomings of current approaches and to discuss more rigorous approaches and protocols. Whether such a discussion will lead to a 'community of practice' (Molinari et al, 2019), professional 'code of practice' guidelines (Doyle et al., 2019), or to quality assurance guidelines remains open. Anyhow, we agree with Goerlandt et al. (2017) regarding the cost-effectiveness and utility of hazard and risk analyses. Even if an assessment cannot be fully validated, it is extremely useful. The understanding gained through a thorough analysis, the awareness of the multifarious ways in which a system can fail, and the insights into the effectiveness of risk reduction measures undoubtedly benefit risk management.

**Code/Data availability**

No data or code was used for this work.

## Author contribution

Conceptualization: BM with support from GB, HK, KS, SV; Writing – original draft: BM; Writing – review and editing: BM, GB, RJ, HK, KS, SV.

## Competing interests

At least one of the (co-)authors is a member of the editorial board of Natural Hazards and Earth System Sciences.

## Acknowledgements

Contributions from the projects ClimXtreme (Module C Impacts - Subproject 5: FLOOD, BMBF, 01LP1903E), SPATE (Space-Time Dynamics of Extreme Floods; DFG, GRK 2043/2, FOR2416) and KAHR (Climate Adaptation, Floods and Resilience: Scientific Monitoring of Reconstruction after the Flood Disaster in RLP and NRW, BMBF, 01LR2102F) are gratefully acknowledged. Thanks to Guan Xiaoxiang for drawing Fig. 1.

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
