# Peer review of "Invited perspectives: Safeguarding the usability and credibility of flood hazard and risk assessments"

_EGUsphere, 2024_

## Author Response (AR1)

**Authors' Reply**

Thanks to both reviewers for their helpful comments. Below we respond to all comments (in blue). The line numbers in blue refer to the revised version with track changes.

**RC1 – Thorsten Wagener**

Merz and colleagues provide an interesting, well supported and relevant discussion of the issue of validation in the context of flood hazard/risk assessments. My comments below hopefully help to clarify some points and to push their discussion just a little bit further. My points are not listed in order of importance.

**[1]** One issue that could be better explained is the meaning of key terminology used. The authors state (line 149…): "Firstly, validation can establish legitimacy but not truth." The relevance of this statement is difficult to understand unless you tell the audience that the actual meaning of the term "validation" is. The main argument of Oreskes (which the authors cite) is that validation translates into something like "establishing the truth". It would be helpful to include this here. The same terminology clarity holds for the term verification that the authors also use.

Our understanding of validation is explained in lines 53 – 80 ("… evaluating its [the model's] ability to achieve its intended purpose. In essence, the evaluation of an FHRA model's validity is determined by its fitness for its intended purpose, reframing the criteria for assessment from correspondence to reality to alignment with decision-making needs …" and "… Our focus on a fit-for-purpose approach follows earlier arguments … the process of structured reasoning about the level of confidence needed to support a particular decision and the credibility of the assessment of risk in that context …". However, we agree that a more explicit definition of the term validation is helpful for the reader. Thus, we have added the following sentence, following Eker et al. (2018) that T. Wagener proposes in his comment [9]:

Line 59: *We thus define validation as "a process of evaluating a model's performance and suitability for its intended use" (Eker et al., 2018).*

Actually, we don't use the term verification in our paper, but verifiability which is explained in Table 2 and Figure 1.

**[2]** A wider point. I find this discussion of "true" in the context of models really unhelpful. A model cannot be true (in my opinion). A model is by definition a simplification of reality. At least for environmental systems, I do not see how there could be a single way to simplify the system reach a "true" model. This is why (if I remember correctly) Oreskes and colleagues suggest using the term evaluation instead of validation. Why did the authors not include this element in their discussion, but chose to assume that validation as a term is what we should continue to use?

On the question whether we should use the term validation: Yes, Oreskes et al. argue that the terms validate and verify are problematic: " … both verify and validate are affirmative terms: They encourage the modeler to claim a positive result . And in many cases, a positive result is presupposed. For example, the first step of validation has been defined by one

group of scientists as developing "a strategy for demonstrating [regulatory] compliance". Such affirmative language is a roadblock to further scrutiny. A neutral language is needed for the evaluation of model performance …" While Oreskes et al. note this problem, they don't really suggest to substitute validation by evaluation.

Evaluation is rather a procedure. Oreskes et al. (1994) mention that the term validation implies legitimacy, e.g., a contract that has not been nullified is a valid one. They mention, however, that this term is often misused by using it in a sense of verification, i.e., consistency with observations, and in a sense that a valid model is a realistic representation of physical reality, i.e., "truth". Our definition of validation as being fit for purpose resonates with the notation by Oreskes et al. in a sense that we agree on a "contract" for model use for a specific decision-making purpose, given certain model properties and quality ensuring procedures, and this makes it legitimate or valid. In addition, the term validation is extremely widespread. The majority of papers discussing the simulation of environmental systems use validation, including those papers that agree that "do not see how there could be a single way to simplify the system reach a "true" model". Thus, to better connect to the existing literature and terms used, we prefer to use validation instead of the evaluation. However, we have added the following sentence as disclaimer that validation has this affirmative touch:

Line 60: *While the term "validation" has been criticised for potentially implying that a model can be established as true, and for encouraging modellers to claim positive results (Oreskes et al., 1994, Wagener et al., 2022), we use the term validation because it is the preferred term for evaluating models in hydrology and water resources management.*

On the discussion of "true": We have checked how we use "true" or "truth" in the manuscript. There are only 3 instances where we use them:

" … Firstly, validation can establish legitimacy but not truth. Truth is unattainable because …" (Line 159) and "… The ideal model-building process utilizes an initial model to make testable predictions, then takes measurements to test and improve it (Ewing et al., 1999). This predictive validation approach appeals because the modeler is unaware of the truth at the time of the model experiment and is therefore not subject to hindsight bias…" (Line 207).

In the revised manuscript, we have kept the first 2 instances, as we basically say what T. Wagener means ("A model cannot be true"). Concerning the third instance, we have substituted "truth" by "measurements". (Line 208).

[3] The authors state (Line 152…): "A model that does not reproduce observed data indicates a flaw in the modelling, but the reverse is not true" Well, no model perfectly reproduces observations (at least not in our field). So, we are generally talking about how well or how poorly as model reproduces observations.

We have deleted this sentence in the revision as we don't need it to make our point, which is that validation can establish legitimacy but not truth.

[4] The authors state (Line 154): "Finally, validation is a matter of degree, a value judgement within a particular decision-making context. Validation therefore constitutes a subjective

process." Yes, I agree that validation is a question of degree. So, validation must contain subjective choices (of thresholds), but does that make the process subjective?

We agree that there is a difference between subjective choices and a subjective process and have reformulated as follows: "Validation therefore includes subjective choices".

[5] In the context of what the authors present, isn't the key question how we decide on appropriate thresholds for the matter of degree of validation? The current discussion does not say much about how we find and agree on these thresholds.

It is indeed an important question how we decide on appropriate thresholds. Here, a basic problem is that the specific thresholds and the ways how to decide on them certainly vary between different contexts. We believe that our framework can help to decide, in a certain context, whether the specific model is valid enough. We follow Howard (2007) who discussed the related problem of what a good decision is. According to Howard, a decision should not be strictly judged by its outcome: A good decision does not always lead to a good outcome and a bad decision does not always lead to a bad outcome. Instead, a good decision is governed by the process that one uses to arrive at a course of action. Howard then defined 6 (decision quality) elements and argued that good decisions are those in which all of these elements are strong. Similarly, we think that our framework can support the discussion about the degree of validation of a model. We think that a good validation is governed by the process that one uses to evaluate a model. Applying our framework (Table 2), we argue that a specific model is validated when all 7 criteria are fulfilled to an extent that is appropriate in the specific context. Of course, the problem still remains what "appropriate in the specific context" means. However, given the large range of contexts where FHRAs are performed, we think that it is not possible to discuss the many ways that parties involved in a FHRA might decide whether a model is valid enough.

In the revised version, we have added a discussion on this question as follows:

Line 428: *These elements are not intended as a recipe to be strictly followed, and our framework does not answer the question of how to decide on the appropriate thresholds that define that a FHRA is "sufficiently valid". This is beyond the scope of this perspective, as the specific thresholds and the ways to decide on them will vary between different contexts. We believe that our framework helps to decide whether the specific model is suitable for a given context. We follow Howard (2007) who discussed the related problem of what constitutes a good decision. According to Howard, a decision should not be judged strictly by its outcome, as a good decision does not always lead to a good outcome, and a bad decision does not always lead to a bad outcome. Instead, a good decision is determined by the process by which one arrives at a course of action. Howard then defined six (decision quality) elements and argued that good decisions are those in which all of these elements are strong. Similarly, our framework can support the discussion of the degree of validation of a model. We believe that good validation is determined by the process used to evaluate a model. Applying our framework, we argue that a particular model is validated when all seven criteria are met to an extent that is appropriate in the specific context.*

[6] One issue is the full acceptance or the complete rejection of models for a specific purpose in the context of model validation in most (all?) studies. Isn't this black and white view a key problem? How do you consider imperfect suitability of models? In current studies

that include some type of validation, there is generally no consideration of the degree in which a model failed to reproduce the observations into future predictions. How would we solve this problem?

Yes, at some point in the process of validating a FHRA, there is typically a decision that the specific model is valid enough (despite its imperfect suitability). And such decisions are required, because a flood protection measure needs to be designed (or any other real-world decision needs to be taken) based on a concrete model output (which can be a single number, a probability distribution, a range of what-if scenarios, etc.). In such a situation, one could consider the degree to which model is valid (its validity) in the decision context. Often, this is already done. For instance, reliability engineering (e.g. Tung, 2011) considers (aleatory and epistemic) uncertainty in the design of structures. In a situation where the available model is less able to reproduce the observations, one can consider this lower validity in a wider probability distribution of the load (external forces or demands) on the system or the resistance (strength, capacity, or supply) of the system. This, in turn, will lead to higher design values due to our high uncertainty represented in the specific model. In a situation where one has several, alternative models, each associated with a measure of how valid they are (e.g. by quantifying their agreement with observations), one can weigh these models to obtain the concrete model output required for a specific decision. We agree with Thorsten Wagener that the full acceptance of models is a key problem and we have added the following discussion on that problem, including some reflections on how one could incorporate the degree of validity in decision-making contexts.

Line: 445: *In the process of model validation, there is usually a decision that the specific model is valid enough (despite its imperfect suitability). Such decisions are needed because a flood protection measure has to be designed, or any other real-world decision has to be made, based on a concrete model output, which can be a single number, a probability distribution, a set of what-if scenarios, etc. In such a situation, one could consider the degree to which the model is valid in the decision context. For instance, reliability engineering (e.g. Tung, 2011) considers (aleatory and epistemic) uncertainty in the design of structures. In a situation where the available model is less able to reproduce the observations, one can consider this lower validity in a wider probability distribution of the load (external forces or demands) on the system or the resistance (strength, capacity, or supply) of the system. This, in turn, will lead to higher design values due to our high uncertainty represented in the specific model. In a situation where one has several alternative models, each associated with a measure of its validity, one can weigh these models to obtain the concrete model output required for a specific decision.*

The authors state that (Line 352) "It is therefore important to state the range for which the model is credible.". However, I do not think this is solving the issue. For once, it still implies that the model is correct if used in the right range, which I think is a very strong assumption.

We think that stating this range (by specifying the range of return periods, failure mechanisms etc., for which data exist, should be specified, as should those cases for which observations are unavailable) is extremely helpful. However, we have clarified in the revised version that we don't assume that the model is correct even when it is valid enough in a specific context:

Line 372: *It is therefore important to state the range for which the model is credible, i.e. for which we assume that the model provides information on the real-world system behaviour that is sufficient for the decision context at hand.*

[7] A key problem for impact of risk models – as far as I know – is the lack of data on flood (or other perils) impact data (e.g. doi.org/10.5194/gmd-14-351-2021). One reason why CAT modelling in practice is so concentrated with a few firms (which own such data). How can we overcome this problem? How can we "validate" without data?

We have mentioned the challenge of the lack of data at several locations in the original manuscript, most prominently in Lines 101 – 109: "… One fundamental problem is that flood risk, i.e. the probability distribution of damage, is not directly or fully observable. Extreme events that lead to damage are rare, and the relevant events may even be unrepeatable, such as the failure of a dam (Hall and Anderson, 2002). The rarity of extreme events results in a situation characterized by both limited data availability and increased data uncertainty. This uncertainty relates to data against which the flood model can be compared. For instance, streamflow gauges often fail during large floods, and losses are not systematically documented and reported losses are highly uncertain. In addition, input data is often insufficient for developing a viable flood model. For example, levee failures depend on highly heterogeneous soil properties, and levee-internal characteristics are typically unknown. Thus Molinari et al. (2019) conclude that "a paucity of observational data is the main constraint to model validation, so that reliability of flood risk models can hardly be assessed…". Our framework is our answer to this challenge. The framework goes beyond the current, most prominent view on model validation (which is strongly focussed on data validation, i.e. comparing simulation against observation) by adding validation elements and criteria (see Figure 1) that can be applied without observations.

In the revised version, we have added the following text:

Line 455: *In summary, our framework is an attempt to address the challenges of FHRA validation, including the need for validation without data. Our framework goes beyond the current, most prominent view of model validation (which is strongly focused on historical data validation, i.e. on comparing simulations and observations) by adding validation elements and criteria (see Figure 1) that can be applied without observations.*

[8] I like the inclusion of Sensitivity Analysis as strategy in Table 1 and in the wider discussion in the paper. Though I do think that its value is wider than discussed here. Wagener et al. (2022) discuss at least four questions that this approach can address in the context of model evaluation (used to avoid the term validation in line with the ideas of Oreskes et al.): (1) Do modeled dominant process controls match our system perception? (2) Is my model's sensitivity to changing forcing as expected? (3) Do modeled decision levers show adequate influence? (4) Can we attribute uncertainty sources throughout the projection horizon?

Thanks a lot for this wider perspective on sensitivity analysis. We added the following text in the revised version:

Line 235: *There are proposals to take a broader view of sensitivity analysis in model evaluation that go beyond modelled input-output relationships (response surfaces). For*

*example, Wagener et al. (2022) argue that (global) sensitivity analysis can provide transparency of model behaviour and increase stakeholder confidence in model results, particularly in data-poor situations, by addressing a range of questions such as: Are the modelled dominant process controls consistent with our perception of the system?*

**[9]** Another interesting reference for the authors might be the study by Eker et al. (2018) who reviewed validation practices. They found, among other things, a total dominance of validation strategies using fit to historical observations (even in the context of climate change studies).

Thanks a lot for this reference, which is highly relevant. We have included this reference in Line 55, when we speak about the dominance of observation-based approaches, and in Line 60, when we define the term validation:

Line 55: *The conventional understanding of model validation prevalent in hydrology and water resources management, but also in the broader field of environmental modelling, entails evaluating the alignment between a model and observed reality, such as streamflow observations (Biondi et al., 2012, Eker et al., 2018).*

Line 60: *We thus define validation as "a process of evaluating a model's performance and suitability for its intended use" (Eker et al., 2018).*

**[10]** Some of the points discussed here are also part of what others have called uncertainty auditing (doi.org/10.5194/hess-27-2523-2023) or sensitivity auditing (doi.org/10.1016/j.futures.2022.103041). These ideas might be interesting to the authors.

Thanks a lot for these references. We have included the toolbox of Page et al. (2023) in Line 230 and have added the idea of sensitivity auditing:

Line 76: *Outside the risk analysis literature, others have also argued for a broader approach to model validation than simply comparing simulation results with observations. For example, Eker et al. (2018) and Wagener et al. (2022) argue for more integrated approaches that also assess the conceptual and methodological validity, and Saltelli and Funtowicz (2014) propose sensitivity auditing, based on a 7-point checklist, to increase the credibility of a model.*

**RC2**

[1] There is almost no consideration given here to the consequences side of risk. Without that, there is a real danger that you are validating hazard rather than risk. You quote Bates (2023) but none of the pre existing research on which that paper is based is credited, fundamentally about comparisons between insurance claims and risk as a modelled. That research gets closest to the proper validation of risk (as opposed to hazard) than much other less comprehensive investigations.

Our paper is not a review on the validation of flood hazard and risk assessments, but a perspective paper which attempts to offer a broader view on validation. To this end, we have tried to cite examples that cover the entire range of flood hazard and risk assessments. However, as validating risk is more difficult than validating hazard, we agree that putting more emphasis on examples that deal with validating the modelling of flood consequences will make the paper more useful. In the revision, we have added a few examples on validating consequences / risk.

Line 194: *One of the rare comparisons of modelled risk, using EAD (Expected Annual Damage) as a proxy for observed risk is based on integrating 20 years of insured losses in the UK (Penning-Rowsell, 2021) the underlying assumption is that the observed EAD integrates a sufficient share of total risk. This comparison finds that modelled flood risk at the national scale is between 2.1 and 9.1 times the corresponding flood loss measured in terms of the insurance compensation paid. In contrast, Bates et al. (2023) find a very good agreement (difference of 2%) between the simulated EAD for 2020 for UK with the observed value reported by the Association of British Insurers. Sairam et al. (2021) compare the simulated damage for large-scale flood events in Germany between 1990 and 2003 with reported damage; for four out of the five events, the uncertainty bounds encompass the reported damage. The damage of the event in 2002 is substantially underestimated by the model, which can be explained by the more than 100 dike breaches not considered in the model.*

Also, insufficient attention is given to the biases involved in flood risk assessment being undertaken by those who benefit from large risk numbers. Many of those developing risk models are engineers intimately concerned with projects to construct flood risk reduction measures, hence the tendency for exaggeration by the models in comparison with real world data on actual flood impacts. Indeed, it seems that a review of most models show widespread exaggerations over anticipated consequences.

The point that engineers tend to exaggerate flood risk in order to benefit from large risk numbers is highly interesting. However, we could not find evidence for the statement of the reviewer "… Indeed, it seems that a review of most models show widespread exaggerations over anticipated consequences …". There are very few papers that point in this direction. One example is 'Penning-Rowsell, E. C. (2021). Comparing the scale of modelled and recorded current flood risk: Results from England. Jour. Flood Risk Manag'. It compares the modelled numbers of flood risk for England with loss figures quantified in terms of insurance claims data and finds that modelled results are between 2.06 and over 9.0 times the comparable flood losses measured in terms of the compensation paid. Although Penning-Rowsell finds this overestimation, he says: "The reasons for these differences remain unclear", and his review of possible reasons does not mention an exaggeration in order to

benefit from large risk numbers. While we agree that the danger of exaggeration exists, ethical standards of the engineering profession and the public and political scrutiny of large infrastructure projects serve as checks against such behaviour. In many regions/countries, there is a strong emphasis on demonstrating the effectiveness and cost-benefit ratio of flood protection measures, which suggests a focus on justifying any proposed measures rather than exaggerating risks or promoting unnecessary construction. Given the lack of evidence on this point in the literature, we decided to forego to discuss this aspect in our manuscript.

**[3]** The part of the paper on Emergency Management is not very convincing. It is very brief and the claims for its advancement over current practises cannot easily be verified. So what are the improvements, and how do they come about? Indeed, the final paragraph of that section rather implies that modelling extreme events in real time is not likely to be credible. That raises the question about the credibility of models generally, rather than simply a discussion of the way that the results can be validated. It also suggests this is rather a bad example.

We don't understand the statement: "… Indeed, the final paragraph of that section rather implies that modelling extreme events in real time is not likely to be credible…". The final paragraph is "… The use case of emergency flood management exemplified in Table 3 reflects the situation that flood models for the management of extraordinary situations cannot rely on typical elements of validation, such as comparing model simulations with observed data. Only rarely does observed data about inundation, defence failures, and impacts exist for a particular region. This is precisely why it is all the more important to safeguard the usability and credibility of the models applied…" Why does this paragraph imply that modelling in real time is not credible? We think that models can be useful and credible even when we don't have (many) observations.

We have chosen the example of emergency management deliberately because this is an area without much data to validate models. We think that our framework is helpful exactly in such situations, when there is little data and when the typical approach of validation (compare against observations) is not possible.

We have added the following sentences to better convey why we think our framework is useful for this case:

Line: 464: We thus focus on the sector that is furthest away from flood modelling and comprehensive FHRAs *and is strongly challenged by the need to validation without data.*

We have extended the last paragraph as follows:

Line 502: The use case of emergency flood management exemplified in Table 3 reflects the situation that flood models for the management of extraordinary emergency situations cannot rely on the typical element of validation, namely comparing model simulations with observed data. Only rarely does observed data about inundation, defence failures, and impacts exist for a particular region. This is precisely why it is all the more important to safeguard the usability and credibility of the models applied *by complementing the traditional approach with a largely observation-independent approach. The latter includes elements that ensure a robust validation procedure (level 1 of the framework in Table 1: procedure-based), so that the assessment is unbiased, transparent, consistent with the*

*existing knowledge and relevant to the decision context. It also includes elements that ensure that the sensitivity and uncertainty of the model are fully understood, including the potential occurrence of major errors (level 2: outcome-based). Finally, it tries to clarify how serious possible errors are (level 3: impact-based). Such an approach could, for example, lead to a decision to evacuate people even though the probability of life-threatening flooding is low given the uncertainty in the current flood forecast.*